

# Reweighting Monte Carlo predictions and automated fragmentation variations in Pythia 8

Christan Bierlich[1★], Philip Ilten[2†], Tony Menzo[2‡], Stephen Mrenna[2,3∘],
Manuel Szewc[2§], Michael K. Wilkinson[2¶], Ahmed Youssef[2∥] and Jure Zupan[2⊥]

**1** Department of Physics, Lund University, Box 118, SE-221 00 Lund, Sweden
**2** Department of Physics, University of Cincinnati, Cincinnati, Ohio 45221, USA
**3** Scientific Computing Division, Fermilab, Batavia, Illinois, USA

★ christian.bierlich@hep.lu.se ,   † philten@cern.ch ,   ‡ menzoad@mail.uc.edu ,
∘ mrenna@fnal.gov ,   § szewcml@ucmail.uc.edu ,   ¶ michael.wilkinson@uc.edu ,
∥ youssead@ucmail.uc.edu ,   ⊥ zupanje@ucmail.uc.ed

## Abstract

This work reports on a method for uncertainty estimation in simulated collider-event predictions. The method is based on a Monte Carlo-veto algorithm, and extends previous work on uncertainty estimates in parton showers by including uncertainty estimates for the Lund string-fragmentation model. This method is advantageous from the perspective of simulation costs: a single ensemble of generated events can be reinterpreted as though it was obtained using a different set of input parameters, where each event now is accompanied with a corresponding weight. This allows for a robust exploration of the uncertainties arising from the choice of input model parameters, without the need to rerun full simulation pipelines for each input parameter choice. Such explorations are important when determining the sensitivities of precision physics measurements. Accompanying code is available at **gitlab.com/uchep/mlhad-weights-validation**.

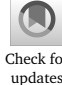

# 1 Introduction

Almost all collider tests of the Standard Model (SM) of particle physics rely on predictions obtained using event generators [1, 2]. An important part of these tests is the estimation of uncertainties on those predictions, which can often be obtained by varying input parameters to the event generator. An important and practical consideration is the efficiency of the algorithms used for uncertainty estimations. Already, efficient reweighting methods exist for the hard process and the parton shower [3–7] and various weighting techniques are used to match and merge the hard process and parton shower, although typically not in the context of variations [8–20]. Similar methods for estimating the uncertainties in hadronization have, up to now, remained elusive. The standard procedure to handle hadronization uncertainties, prior to this work, was to perform repeated simulations with different sets of values for the relevant hadronization model's parameters, where the values are chosen such that the model's predictions remain compatible with the reference data [21–23]. In this manuscript, we remedy this by introducing an efficient solution for reweighting kinematic parameters of hadronization and provide an implementation for the complete hadronization model of the PYTHIA 8 Monte Carlo event generator.

The uncertainties in the prediction for the hard process, based on matrix elements, are typically estimated by varying the factorization and renormalization scales, although this does not capture, of course, our full ignorance of the importance of missing, higher-order terms in the calculation. Additional uncertainties in hard-process calculations can arise from the choice of couplings and the parton distribution functions, including the scale at which they are evaluated. An event weight can then be calculated from the ratio of the hard process calculated with the modified choices to the baseline ones. For parton shower uncertainties, the situation is more complicated because the probability distributions must be evaluated many times and must preserve unitarity. Here, a modification of the Monte-Carlo sampling algorithm is necessary to account correctly for accepted-and-rejected trial emissions.

The standard procedure to handle hadronization uncertainties is to perform repeated simulations with different sets of values for the relevant hadronization model's parameters, where the values are chosen such that the model's predictions remain compatible with the reference data. Statistical comparisons can then be made on observables relevant to a particular analysis. While this is straightforward, it is also computationally expensive, especially if the predictions are further simulated at the detector level (material interactions, detector response, *etc*.). It is advantageous to use instead only one sample of events in the detector simulation, and then compute relative probabilities for different hadronization parameter choices.

Conceptually, it is not difficult to calculate an alternative probability for a given accepted hypothesis. In practice, it can be technically challenging to re-organize an existing Monte Carlo method to do so. In this paper, we describe a method to calculate relative probabilities for predictions based on a veto algorithm and apply it to uncertainty estimates in simulations of hadronization based on the Lund string fragmentation model [24]. The presented method is similar to the one used previously for parton shower uncertainty estimates [3, 5–7, 15], and is implemented here for kinematic-based parameters of the string model, not flavor-based parameters.[1] A key difference between the parton shower and hadronization is that the parton shower uses a veto algorithm; a no-emission probability up to a given scale $Q$ is calculated, and then an emission is produced at that scale. For hadronization, the scale of the next emission is always at the scale determined from the previous emission. While both processes are Markov chains, this distinction critically changes how the varied parameter uncertainties are propagated.

---

[1]The flavor selector in PYTHIA 8 does not use the same accept-reject algorithm as for kinematics, and is as such an entirely separate challenge.

While this manuscript provides an efficient method to compute fragmentation uncertainties in PYTHIA 8 specifically, it is also applicable more broadly. It could, for instance, be applied to cluster hadronization-models [25–31]; various machine-learning based hadronization-models, such as those described in refs. [32–34]; or the multiparton interaction model within PYTHIA 8 itself [35]. For instance, because the multiparton interaction (MPI) model in PYTHIA 8 produces additional strings in a parton-shower-like fashion, the method presented here is already directly applicable to the variation of MPI parameters. Furthermore, the variation of kinematic hadronization parameters, described here, already works "as is" when the MPI model is switched on. However, it is important to note that as the number of interactions per event increases, the spread in the produced weights will also increase, and so care must be taken to ensure the reweighted sample has sufficient statistical power.

The paper is organized as follows: In section 2, we provide a detailed presentation of the proposed method for fast uncertainty estimation in hadronization simulations. Then, in section 3, we validate the effectiveness of the method by applying it to two distinct data samples. Finally, in section 4, we summarize our findings and draw conclusions.

## 2 Method

An event produced by an event generator, like PYTHIA 8, begins from a small number of partons that evolve through various stages. At each stage the color quantum numbers are tracked in the large color $N_c$ limit, such that each new color is assigned a new color index. In this limit, only planar color flows are retained, and colored partons can be assigned a unique pair of integers to represent color and anticolor. After the perturbatively-motivated evolution of the parton shower, one of the last stages in the event development is hadronization. Prior to this step, the collection of quarks, antiquarks, and gluons can be partitioned into color-singlet objects (strings) based on their color quantum numbers. The Lund string model of hadronization [24, 36, 37] is then applied to reduce strings into the observed hadrons. The string represents a flux tube of the non-perturbative strong force between a quark and an antiquark that successively breaks into hadrons, represented by stable oscillating string states characterized by their four-momentum $p_h$ and flavor. The full probability of a given fragmentation can be split into a flavor selection, a transverse momentum sampling, and a longitudinal momentum sampling, which are all combined to ensure a physical emission. A detailed discussion of the Lund fragmentation function as implemented in PYTHIA 8 can be found in ref. [38]. Here, we summarize those elements needed for the uncertainty estimation of the hadronization.

The Lund symmetric fragmentation function, or scaling function, determines the probability for a hadron to be emitted with longitudinal lightcone momentum[2] fraction $z$. The momentum fraction is defined as the fraction of the *remaining* lightcone momentum taken away by the hadron, with the remaining $1-z$ lightcone momentum fraction available for the subsequent production of hadrons. The fragmentation function has the following form:

$$f(z) \propto \frac{1}{z^{1+r_Q b m_Q^2}} (1-z)^a \exp\left(-\frac{b m_\perp^2}{z}\right), \tag{1}$$

where $Q$ is the quark flavor, $m_Q$ is the quark mass, $m_\perp^2 \equiv m^2 + p_T^2$ is the square of the transverse mass, where $m$ is the hadron mass and $p_T$ is the transverse momentum of the hadron, and $r_Q$, $a$, and $b$ are the constant parameters fixed by fits to experimental data.[3] The Bowler modification

---

[2]The lightcone momentum is defined as $E \pm p_z$ for a parton moving in the $\pm z$ direction.

[3]The default parameter names and values as implemented in PYTHIA 8 are `StringZ:aLund = 0.68`, `StringZ:bLund = 0.98`, `StringZ:rFactC = 0`, and `StringZ:rFactB = 0.855` for $a$, $b$, $r_c$, and $r_b$, respectively.

$z^{-r_Q b m_Q^2}$ in eq. (1) is only included for heavy quarks, *i.e.*, $r_Q = 0$ unless $Q \in \{c, b\}$ [39]. PYTHIA 8 also allows for modifications to the $a$-parameter to be used in splittings involving strange quarks $s$ or diquarks $D$, parameterized by the form $a_i' = a + \delta a_i$, where $\delta a_i$ represents an adjustable parameter[4] within PYTHIA 8 with $i \in \{s, D\}$; the form of $f(z)$ is also modified from eq. (1), accounting for the fact that the emitted quarks can be of a different flavor than the endpoints of the original string. The maximum of $f(z)$, denoted $f_{\max}$, can be determined analytically for a given set of input parameter values, denoted $c_i$. Sampling $z$ from $f(z)$ is done by selecting a pseudo-random number $x$ until one satisfies $x < f(z)/f_{\max} \leq 1$, a method known as the accept-reject algorithm, further described in section 2.1.

In the default model of PYTHIA 8, the transverse momentum $p_T$ of each emitted hadron is obtained by first generating a back-to-back transverse momentum of each new $q\bar{q}$ pair string break, sampling from a Gaussian distribution. The physics origin of the Gaussian is the calculation of the tunneling probability of the $q\bar{q}$ pair, through a classically forbidden region, calculable in the WKB approximation as [40]

$$\frac{\mathrm{d}\mathcal{P}}{\mathrm{d}^2 p_T} \propto \exp\left(-\pi m_q^2/\kappa\right) \exp\left(-\pi p_T^2/\kappa\right), \tag{2}$$

where $\kappa$ is the string tension, and $m_q$ and $p_T$ are the quark masses and transverse momenta respectively. The quark mass (and thus flavor) and $p_T$ can therefore be generated separately. In this paper we will introduce reweighting for the parameter controlling the $p_T$ distribution, and address the reweighting of the flavor parameters in a future paper.

The $q$ from a particular string break combines with a $\bar{q}$ from an adjacent string break to produce a hadron. In principle, the generation of the hadron $p_T$ should be parameter free, with $\kappa$ known, based on the arguments above. In practice, however, generating Gaussian $p_T$ kicks with $\sigma_{p_T}^2 = \kappa/\pi \approx (0.25\ \text{GeV})^2$ produces too soft hadron spectra. Therefore, $\sigma_{p_T}$ is left as a free parameter.[5] The end effect is that the hadron $p_T$ is composed of the quark pair $p_T$, each generated from a Gaussian distribution:

$$P(p_x, p_y, \sigma_{p_T}) = \frac{1}{2\pi\sigma_{p_T}^2} \exp\left[-(p_x^2 + p_y^2)/(2\sigma_{p_T}^2)\right]. \tag{3}$$

Such Gaussian distributions can be sampled with complete efficiency, *e.g.*, using the Box–Muller transform [41].

Our key interest is to calculate uncertainties arising from different choices of the parameters $a$, $a_s'$, $a_D'$, $b$, $r_c$, $r_b$, and $\sigma_{p_T}$ as they enter into eqs. (1) and (3). In the following, we first review the accept-reject algorithm so as to later introduce a modified version of it, best suited for the uncertainty estimation on the parameters of eq. (1). We also explain how to perform uncertainty estimation for $\sigma_{p_T}$ by taking advantage of the direct sampling from eq. (3).

It should be noted that the hadronization algorithm described above is used while the mass of the remaining string is sufficiently large, such that suitable phase space exists to produce a hadron and a remaining string. When the remaining string reaches a sufficiently low mass, a specialized splitting is performed where two hadrons are produced without a remaining string, rather than a hadron and the remaining string [42]. However, this splitting is not always successful; if the remaining string has an $m_\perp$ smaller than the summed $m_\perp$ of the two hadrons, then the entire hadronization of the string is rejected and started over. This rejection only depends upon the kinematics of the string, and does not directly depend upon the kinematic hadronization parameters; the $b$ parameter is used to break the symmetry of the system, but does not change the rejection rate. Consequently, this final possible rejection

---

[4]The default parameter names and values as implemented in PYTHIA 8 are `StringZ:aExtraSQuark = 0` and `StringZ:aExtraDiquark = 0.97`, for $s$ and $D$ respectively.

[5]Within PYTHIA 8, $\sigma_{p_T}$ is set with the parameter name `StringPT:sigma`.

is accounted for by including the weight of all rejected strings, in addition to the weights of the accepted strings.

## 2.1 Standard accept-reject algorithm

The accept-reject algorithm can be used to sample a probability distribution when the maximum value of the probability distribution, or a reliable overestimate thereof, is known. The algorithm for sampling the probability distribution $P(z, c_i)$ begins by defining an acceptance probability $P_{\text{accept}}(z, c_i)$ for a trial value of $z$,

$$P_{\text{accept}}(z, c_i) \equiv \frac{P(z, c_i)}{\widehat{P}} \leq 1\,. \tag{4}$$

Both the acceptance probability $P_{\text{accept}}(z, c_i)$ and the probability distribution $P(z, c_i)$ depend on a set of parameter values $c_i$, that we will later vary. The constant $\widehat{P}$ is chosen so that the relation in eq. (4) is satisfied; it can be either the analytic maximum or a numerically estimated overestimate. A trial value for $z$ is accepted only if $P_{\text{accept}}$ is larger than a random uniform variate. If the trial value of $z$ is rejected, with probability $P_{\text{reject}} = 1 - P_{\text{accept}}$, a new trial $z$ is then selected. The algorithm continues until a given $z$ value is accepted. That is, in the standard accept-reject algorithm, the value of $z$ is selected with probability $p$ given by the product of the final accept probability times a factor accounting for all of the rejected trials:

$$p(z) = P_{\text{accept}}(z) \sum_{n=0}^{\infty} A^n\,, \quad \text{where} \quad A = \int_0^1 \mathrm{d}z' \left(1 - P_{\text{accept}}(z')\right)\,, \tag{5}$$

where the dependence on the chosen parameter values $c_i$ has been suppressed for brevity. Summing the geometric series in $A$ gives,

$$p(z) = \frac{P_{\text{accept}}(z)}{1 - A} = \frac{P_{\text{accept}}(z)}{\int_0^1 \mathrm{d}z'\, P_{\text{accept}}(z')} = P(z)\,, \tag{6}$$

showing that the algorithm yields the desired distribution. The exact value of $\widehat{P}$, provided that $P_{\text{accept}} \leq 1$, only affects the efficiency of the algorithm; the further $\widehat{P}$ is from the actual maximum of $P(z, c_i)$, the less efficient the sampling.

## 2.2 Modified accept-reject algorithm

Next, we present a modification of the accept-reject algorithm that assigns appropriate weights to the existing event, depending on how the parameter values $c_i$ are varied. We refer to the original set of parameter values $c_i$ as the baseline and the new set $c_i'$ as the alternative. If the event generated with the baseline parameters has weight $w$ (typically in PYTHIA 8, $w = 1$), the modified accept-reject algorithm calculates the weight $w'$ that corresponds to the alternative values of the parameters. If $w' > w$, the event is more probable given the alternative parameter values; if $w' < w$, it is less probable.

For the calculation of the weight $w'$, one needs to keep track of all the trial $z$ values in the standard accept-reject algorithm. For each $z$ that was rejected, $w$ is multiplied by $R'_{\text{reject}}(z)$, while for the accepted value of $z$, the multiplication is by $R'_{\text{accept}}(z)$. Here, $R'_{\text{accept}}(z)$ is the ratio of alternative and baseline acceptance probabilities,

$$R'_{\text{accept}}(z) = \frac{P'_{\text{accept}}(z)}{P_{\text{accept}}(z)} = \frac{P'(z)}{P(z)}\,, \quad \text{with} \quad P'_{\text{accept}}(z, c_i') = \frac{P'(z, c_i')}{\widehat{P}}\,, \tag{7}$$

while $R'_{\text{reject}}(z)$ is the ratio of the alternative and the baseline rejection probabilities,

$$R'_{\text{reject}}(z) = \frac{P'_{\text{reject}}(z)}{P_{\text{reject}}(z)} = \frac{1 - P'_{\text{accept}}(z)}{1 - P_{\text{accept}}(z)} = \frac{\widehat{P} - P'(z)}{\widehat{P} - P(z)} . \tag{8}$$

The value of $\widehat{P}$ can always be chosen such that both $P'_{\text{accept}} \leq 1$ and $P_{\text{accept}} \leq 1$, albeit at some loss of efficiency when the equality does not hold for the latter. Explicitly, we can write the per-event hadronization weight as

$$w' = w \prod_{i \in \text{accepted}} R'_{i,\text{accept}}(z) \prod_{j \in \text{rejected}} R'_{j,\text{reject}}(z) , \tag{9}$$

where $w$ is the baseline event weight, the first product is over accepted trials of $z$, and the second product is over the rejected trials of $z$.

We can readily show that the weight $w'$ corresponds to the correct probability $p'(z)$ for selecting the final trial-$z$ value using the alternative parameter values $c'_i$:

$$p'(z) = P_{\text{accept}}(z) R'_{\text{accept}}(z) \sum_{n=0}^{\infty} A'^n , \quad \text{where} \quad A' = \int_0^1 dz' \left(1 - P_{\text{accept}}(z')\right) R'_{\text{reject}}(z') . \tag{10}$$

Summing the geometric series in $A'$ gives

$$p'(z) = \frac{P'_{\text{accept}}(z)}{1 - A'} = \frac{P'_{\text{accept}}(z)}{\displaystyle\int_0^1 dz' \, P'_{\text{accept}}(z')} = P'(z) , \tag{11}$$

as desired.

A few considerations are worth mentioning. As in the case of parton-shower variations, the modified rejection ratio in eq. (8) is inversely proportional to the difference $\hat{P} - P$ and can become large if $\hat{P} \simeq P$, leading to large weights. It is thus advantageous for $\hat{P}$ to not approximate the maximum value of $P(z, c_i)$ too closely, but to be larger by an $\mathcal{O}(1)$ factor. In practice, multiplying $\hat{P}$ by a factor of ten typically leads to stable results.[6] The final event weight $w'$ can also become large in cases when the baseline and alternative probability distributions have limited overlap, *i.e.*, the baseline distribution does not provide proper support for the alternative distribution. Good indicators of the fidelity of the reweighting are the mean weight

$$\mu \equiv \sum_{i=1}^{N} \frac{w'_i}{N} \tag{12}$$

(or, equivalently, the weight sum $\sum_i w'_i$), and the effective number of events

$$n_{\text{eff}} \equiv \frac{\left(\sum_{i=1}^{N} w'_i\right)^2}{\sum_{i=1}^{N} w'^2_i} , \tag{13}$$

where $N$ is the number of generated events. If the mean event weight is not near unity, or if the effective number of events is significantly lower than the actual number of simulated events, care should be taken when interpreting the weighted results.

---

[6]This factor may be adjusted within PYTHIA 8 by modifying the corresponding `overSample` parameter for each alternative parameter, *e.g.*, for parton-shower variations, `UncertaintyBands:overSampleFSR` specifies the over-sample factor for QCD final-state radiation enabled by the `fsr:*` set of variation keywords.

### 2.3 Variation details

Currently, we have implemented variations for the $a, b, r_c$, and $r_b$ parameters of the Lund string fragmentation function $f(z)$ given by eq. (1), and the hadron transverse momentum $\sigma_{p_T}$ of eq. (3). The variation weight for one selection of $\sigma_{p_T}$ does *not* require the use of the accept-reject algorithm but can be calculated directly using the Box–Muller transform:

$$w' = \frac{\sigma^2}{\sigma'^2} \exp\left(-\kappa\left(\frac{\sigma^2}{\sigma'^2} - 1\right)\right), \tag{14}$$

where $\kappa = (n_1^2 + n_2^2)/2$ and $n_i$ are normally distributed random variates.

The two event weights arising from variations in eqs. (1) and (3) can be combined into a single event weight by multiplication, due to the fact that we are sampling in a sequential manner from $P(p_x, p_y)$ and $P(z|p_x, p_y)$, *i.e.*, $P(p_x, p_y)$ does not depend upon $z$. However, variations of the parameters of $f(z)$ must be considered as a group. While a variation of the $a$ parameter for a fixed $b$ parameter can be calculated and *vice versa*, the product of weights from these two calculations is not equivalent to varying both $a$ and $b$ simultaneously. One reason for this is that the maximum weight $f_{\max}(a_1, b_1)$ is different from the maximum weights $f_{\max}(a_1, b_0)$ and $f_{\max}(a_0, b_1)$. This applies to all of the parameters that enter into eq. (1): $a, b, r_c$, and $r_b$.

## 3 Validation

The goal of the presented reweighting method is to enable the use of alternative event weights $w'$ to produce the desired distributions using the original sample of events, rather than generating a new sample for each alternative parameter value. Therefore, we validate the method by generating samples of $10^6$ events using PYTHIA 8 configured with a set of baseline parameter values. During this generation, we also calculate, using the modified accept-reject algorithm, a per-event weight $w'$ corresponding to an alternative set of parameter values. We then compare the $w'$-weighted distributions to those obtained by generating new samples using PYTHIA 8 configured with the alternative parameter values as the baseline and without using the modified accept-reject algorithm.

We do this for the Lund parameters $a$, $b$, and $r_b$, as well as the fragmentation transverse-momentum width $\sigma_{p_T}$.[7] The top panels in figs. 1 to 4 show that the observables of interest, described in further detail below, are sensitive to changes in $a$, $b$, $r_b$, and $\sigma_{p_T}$, respectively. The bottom panels in figs. 1 to 4 show the agreement between the $w'$-weighted distributions and those generated with the alternative values set as the baseline. We also vary parameters $a$ and $b$ simultaneously, and fig. 5 shows the analogous plots for these cases. Whenever not explicitly stated, the parameters are set to their default values of $a = 0.68$, $b = 0.98$, $r_b = 0.855$, and $\sigma_{p_T} = 0.350$ from the Monash tune [43].

### 3.1 Validation simulations

The event samples were all generated using a modified version of PYTHIA 8.310 [44]. For event samples in which $a$, $b$, or $\sigma_{p_T}$ were varied, we simulated electron-positron collisions with a center-of-mass energy at the measured $Z$-boson mass.[8] We then applied the selections

---

[7]Although validated, the reweighting method for the $a'_s$ and $a'_D$ parameters is not shown within this paper for the sake of brevity.

[8]The relevant configuration parameters are `Beams:idA = 11`, `Beams:idB = -11`, `Beams:eCM = 91.189`, `PDF:lepton = off`, `WeakSingleBoson:ffbar2gmZ = on`, `23:onMode = off`, and `23:onIfAny = 1 2 3 4`.

from the ALEPH analysis described in ref. [45] using the corresponding RIVET analysis [46], finally obtaining a dataset consisting predominantly of $Z$-boson decays to hadrons. For validating the variations in $r_b$, we instead simulated proton-proton collisions with a center-of-mass energy of 13 TeV and applied the selections from the LHCb analysis described in ref. [47].[9] These requirements provide a sample of jets that contain a $J/\psi$, have transverse momentum $p_T(\text{jet}) > 20$ GeV, and lie in the pseudorapidity range $2.5 < \eta(\text{jet}) < 4.0$.

In general, the $w'$-weighted distributions are in good agreement with the distributions where the parameter values were set as the baseline. This agreement breaks down if the Lund fragmentation function for the alternative parameter values is large in a range where the Lund fragmentation function approaches zero for the baseline parameter values, as shown in fig. 2 (bottom left) and fig. 3 (bottom right). The reweighting then requires large weights and samples the phase space poorly. To illustrate this point, fig. 6 shows distributions of the Lund fragmentation function for different values of $r_b$ and the corresponding distributions of event weights.

Therefore, care must be taken when selecting the baseline value of a parameter to be varied, since the reweighting method may not successfully reproduce the distributions, if the alternative parameter values are too different. This can be checked by calculating the mean event weight $\mu$, as defined in eq. (12). In the limit of infinite data, and because the events were generated according to $P(z, c_i)$, we can write

$$\mu \equiv \sum_{i=1}^{N} w'_i / N \approx \int d\mathcal{E}\, P(\mathcal{E}, c_i) w'(\mathcal{E}), \tag{15}$$

where $\mathcal{E}$ is an event composed of a series of accepted and rejected values of $z$, with joint probability $P(\mathcal{E}, c_i)$ depending on the generation baseline parameters. Explicitly, and with some abuse of notation, we can integrate over the variable-length sets of accepted and rejected values $\vec{z}_{\text{accepted}}$ and $\vec{z}_{\text{rejected}}$,

$$\mu = \int d\vec{z}_{\text{accepted}}\, d\vec{z}_{\text{rejected}}\, w'(\vec{z}_{\text{accepted}}, \vec{z}_{\text{rejected}}) \prod_{i \in \vec{z}_{\text{accepted}}} P_{i,\text{accept}} \prod_{j \in \vec{z}_{\text{rejected}}} P_{j,\text{reject}}. \tag{16}$$

Introducing the expression for $w'$ in eq. (9), the $P_{i,\text{accept}}$ and $P_{j,\text{reject}}$ factors cancel out and

$$\mu = \int d\vec{z}_{\text{accepted}}\, d\vec{z}_{\text{rejected}} \prod_{i \in \vec{z}_{\text{accepted}}} P'_{i,\text{accept}} \prod_{j \in \vec{z}_{\text{rejected}}} P'_{j,\text{reject}} = \int d\mathcal{E}\, P'(\mathcal{E}, c'_i) = 1. \tag{17}$$

Thus, if the generated events cover appropriately the phase space for both $P(z, c_i)$ and $P'(z, c'_i)$, the weights have an expectation value $\mu$ consistent with one. If $\mu \not\approx 1$, the reweighting method is unlikely to reproduce the distributions well. This can be caused by the baseline distribution providing insufficient coverage for the alternative distribution, since generated datasets are limited by finite statistics.

In addition to $1 - \mu$, the ratio of the effective sample size $n_{\text{eff}}$ (defined in eq. (13)) to the number of generated events $N$ (equivalent to the square of the ratio of the relative statistical uncertainty of the unweighted sample to that of the weighted sample) is also a useful metric, as it describes the statistical power of the reweighted alternate distribution. If $n_{\text{eff}}/N \ll 1$, it may be necessary to adjust the baseline distribution to be closer to the alternate distribution. For

---

[9]The relevant configuration parameters are `Beams:idA = 2212`, `Beams:idB = 2212`, `Beams:eCM = 13000`, `PhaseSpace:pTHatMin = 15`, `HardQCD:HardBBbar = on`, and `PartonLevel:MPI = off`. Additionally, for efficient generation, all $b$-hadron decays which do not explicitly contain a $J/\psi$ are switched off using the `ID:onMode` method. Note that this configuration does not include $b$-hadron production from $g \to b\bar{b}$ splittings, but still provides a useful proxy for the distribution.

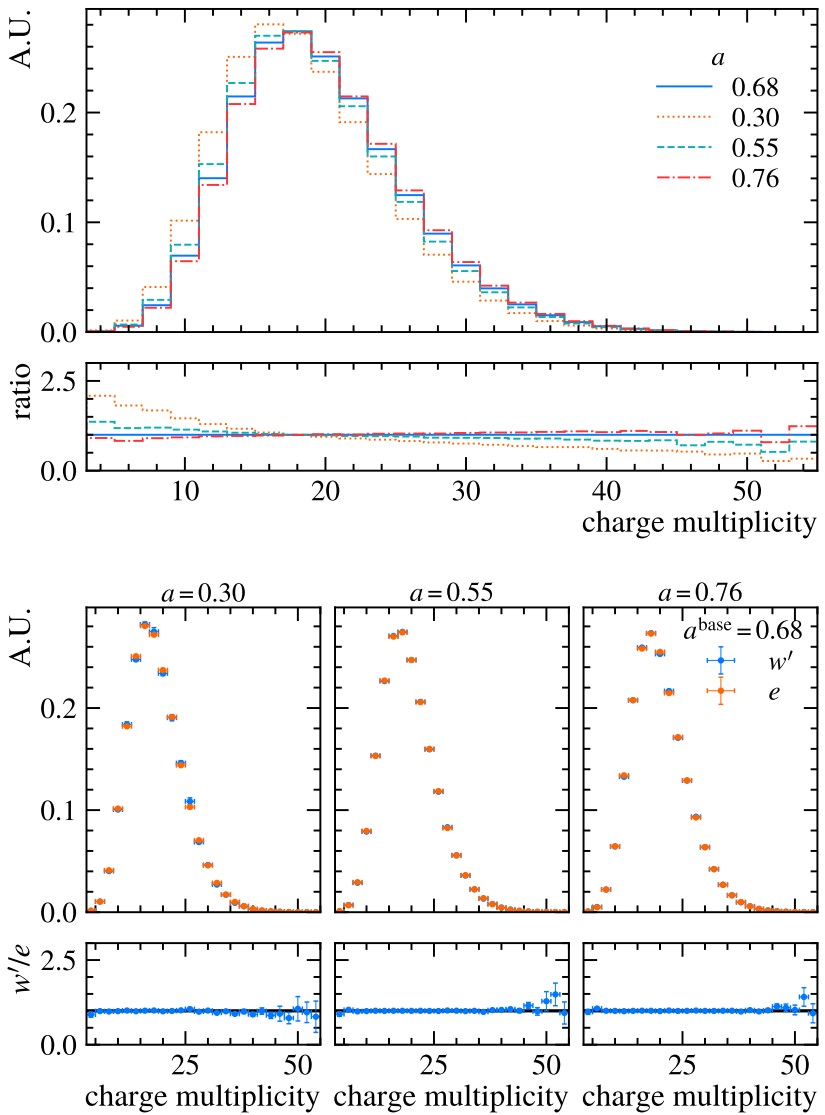

Figure 1: Comparison of the distributions, shown in arbitrary units, of the event charge multiplicity when the parameter $a$ is (top) explicitly set to different values, or (bottom) when it is varied using different methods. In the top panel, the lower row shows the ratios of the distributions generated with various values of $a$ to that generated with $a = 0.68$. In the bottom panel, the distributions labeled $e$ were generated with the value of the parameter $a$ explicitly set to (left) 0.30, (middle) 0.55, and (right) 0.76. The distributions labeled $w'$ are all taken from the same sample generated with $a = a^{base} = 0.68$, but with different sets of alternative event weights, calculated using the accept-reject algorithm applied according to the alternative values of $a$. The bottom row shows the ratios of the latter distributions to the former.

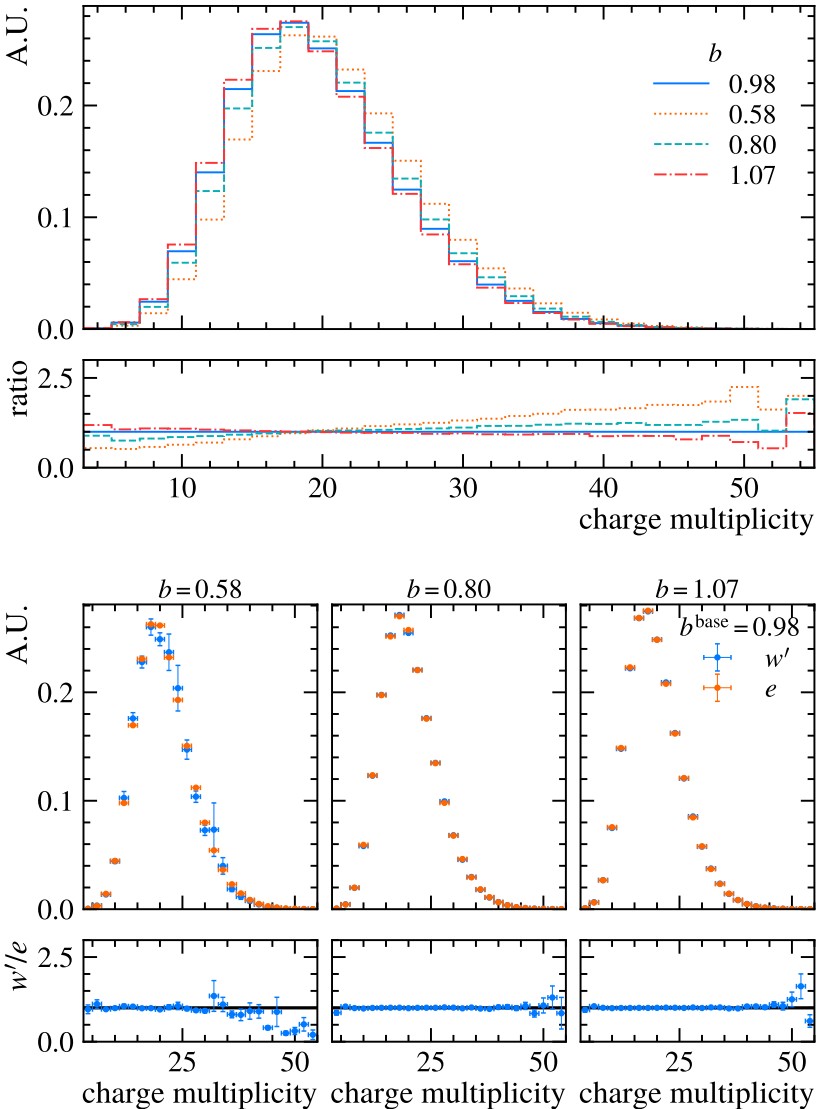

Figure 2: Comparison of the distributions, shown in arbitrary units, of the event charge multiplicity when the parameter $b$ is explicitly set to (top) different values or (bottom) when it is varied using different methods. In the top panel, the lower row shows the ratios of the distributions generated with various values of $b$ to that generated with $b = 0.98$. In the bottom panel, the distributions labeled $e$ were generated with the value of the parameter $b$ explicitly set to (left) 0.58, (middle) 0.80, and (right) 1.07. The distributions labeled $w'$ are all taken from the same sample generated with $b = b^{\text{base}} = 0.98$, but with different sets of alternative event weights, calculated using the accept-reject algorithm applied according to the alternative values of $b$. The bottom row shows the ratios of the latter distributions to the former.

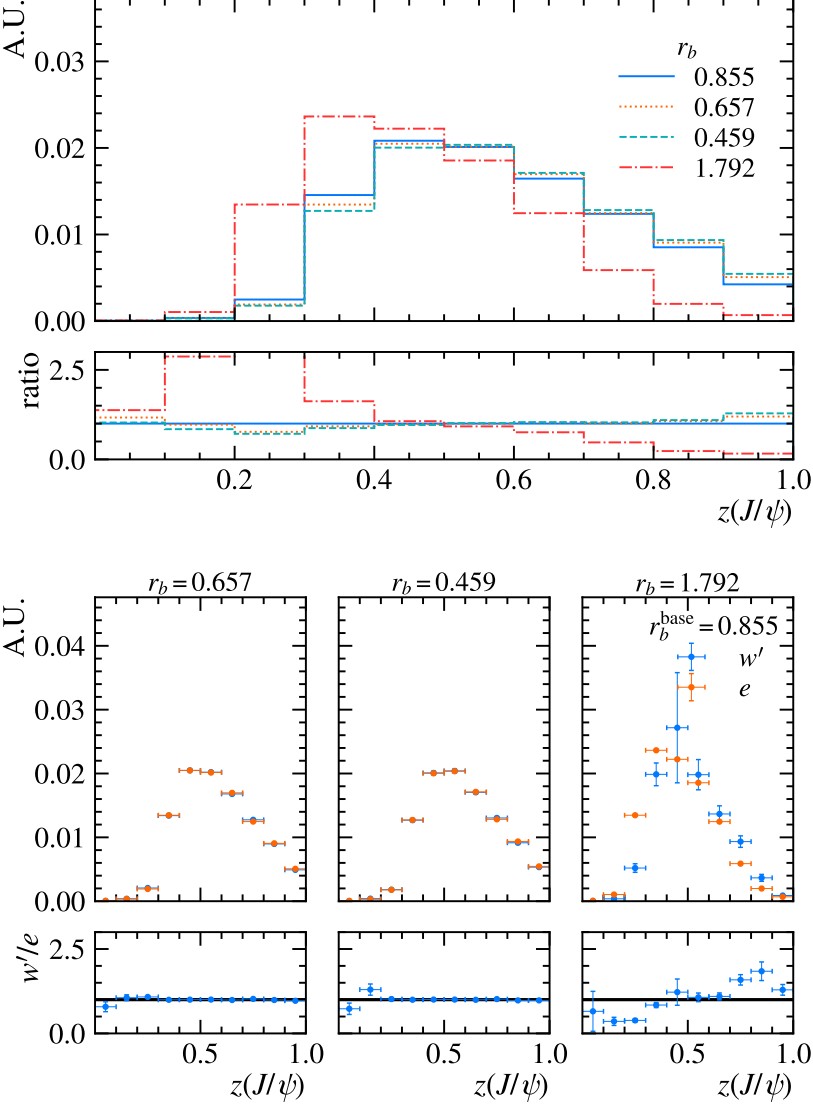

Figure 3: Comparison of the distributions, shown in arbitrary units, of the ratio of the transverse momentum of a $J/\psi$ meson to the transverse momentum of the jet in which it is found, $z(J/\psi)$, when the parameter $r_b$ is (top) explicitly set to different values, or (bottom) when it is varied using different methods; see ref. [47] for details of this analysis. In the top panel, the lower row shows the ratios of the distributions generated with various values of $r_b$ to that generated with $r_b = 0.855$. In the bottom panel, the distributions labeled $e$ were generated with the value of the parameter $r_b$ explicitly set to (left) 0.657, (middle) 0.459, and (right) 1.792. The distributions labeled $w'$ are all taken from the same sample generated with $r_b = r_b^{\text{base}} = 0.855$, but with different sets of alternative event weights, calculated using the accept-reject algorithm applied according to the alternative values of $r_b$. The bottom row shows the ratios of the latter distributions to the former.

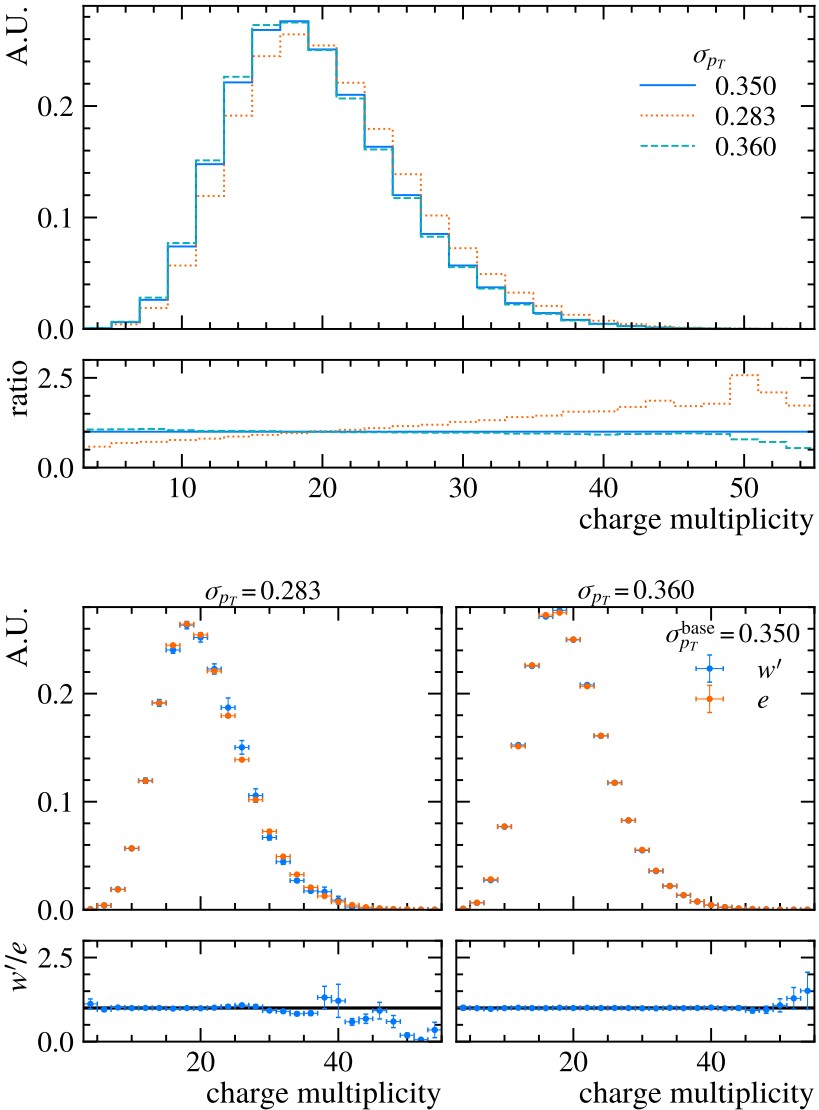

Figure 4: Comparison of the distributions, shown in arbitrary units, of the event charge multiplicity when the parameter $\sigma_{p_T}$ is (top) explicitly set to different values, or (bottom) when the parameter $\sigma_{p_T}$ is varied using different methods. In the top panel, the lower row shows the ratios of the distributions generated with various values of $\sigma_{p_T}$ to that generated with $\sigma_{p_T} = 0.350$. In the bottom panel, the distributions labeled $e$ were generated with the value of the parameter $\sigma_{p_T}$ explicitly set to (left) 0.283 and (right) 0.360. The distributions labeled $w'$ are all taken from the same sample generated with $\sigma_{p_T} = \sigma_{p_T}^{\text{base}} = 0.350$, but with different sets of alternative event weights, calculated using the accept-reject algorithm applied according to the alternative values of $\sigma_{p_T}$. The bottom row shows the ratios of the latter distributions to the former.

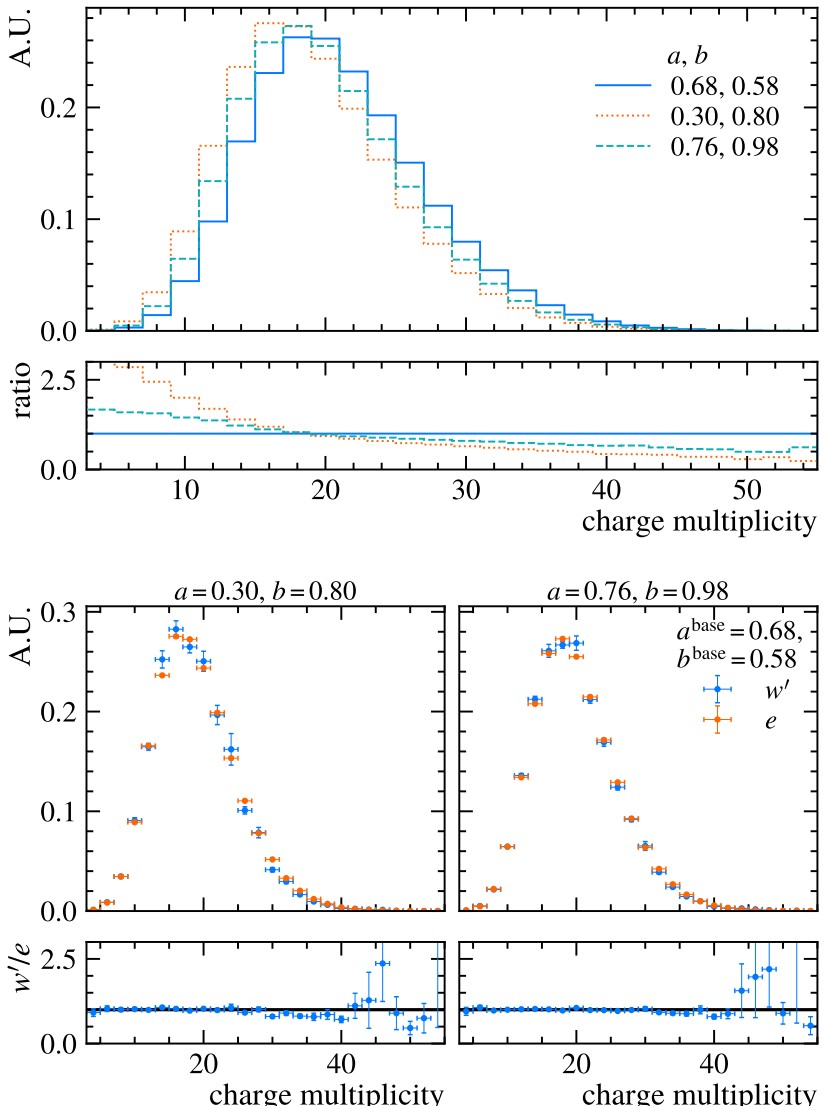

Figure 5: Comparison of the distributions, shown in arbitrary units, of the event charge multiplicity when the parameters $a$ and $b$ are (top) explicitly set to various values, or (bottom) when and $a$ and $b$ are simultaneously varied using different methods. In the top panel, the lower row shows the ratios of the distributions generated with various values of $a$ and $b$ to that generated with $a = 0.68$ and $b = 0.58$. In the bottom panel, the distributions labeled $e$ were generated with the values of the parameters $a$ and $b$ explicitly set to (left) $a, b = 0.30, 0.80$ and (right) $a, b = 0.76, 0.98$. The distributions labeled $w'$ are all taken from the same sample generated with $a = a^{\text{base}} = 0.68$ and $b = b^{\text{base}} = 0.58$, but with different sets of alternative event weights, calculated using the accept-reject algorithm applied according to the alternative values of $a$ and $b$. The bottom row shows the ratios of the latter distributions to the former.

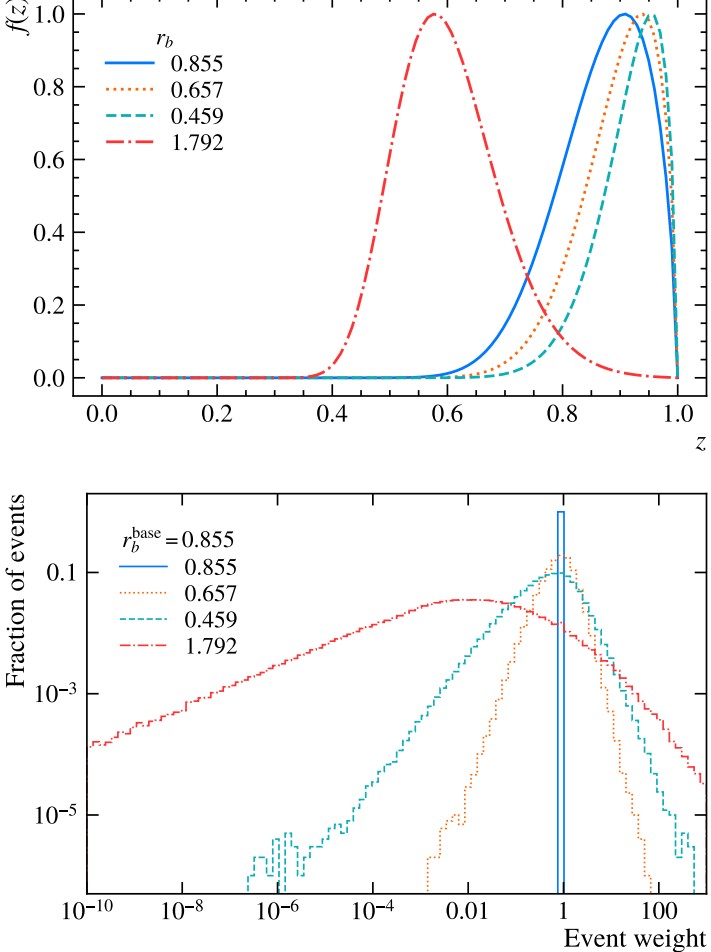

Figure 6: (Top) distributions of the Lund fragmentation function $f(z)$ for different values of $r_b$ with $m_\perp = 25\,\text{GeV}$ and all other parameters set to their default values. (Bottom) distributions of the event weights $w'$ for reweighting from $r_b = r_b^{\text{base}} = 0.855$, shown on a log-log scale with varying bin sizes. Notice that when $r_b = 1.792$, $f(0.575) \approx 1$, but $f(0.575) \approx 0$ for $r_b = 0.855$, resulting in weights far from unity when reweighting from $r_b = r_b^{\text{base}} = 0.855$ to $r_b = 1.792$. The weights when $r_b = r_b^{\text{base}} = 0.855$ are exactly equal to 1, though they may appear shifted due to the binning.

Table 1: Difference of mean weight $\mu$ from one and the ratio of the effective sample size $n_{\text{eff}}$ to the number of generated events $N$ for the listed variations; see the text. A value of zero or one indicates that $\mu$ or $n_{\text{eff}}/N$, respectively, is exactly one, corresponding to the base case where all weights are one.

| variation | $1 - \mu$ | $n_{\text{eff}}/N$ | figure |
|---|---|---|---|
| $r_b^{\text{base}} = 0.855$ | 0 | 1 | |
| $r_b = 0.657$ | $(0.1 \pm 7.9) \times 10^{-4}$ | $6.2 \times 10^{-1}$ | figs. 3 and 6 |
| $r_b = 0.459$ | $(1.2 \pm 2.4) \times 10^{-3}$ | $1.9 \times 10^{-1}$ | |
| $r_b = 1.792$ | $(1.1 \pm 0.4) \times 10^{-1}$ | $7.3 \times 10^{-4}$ | |
| $a^{\text{base}} = 0.68$ | 0 | 1 | |
| $a = 0.30$ | $-(0.5 \pm 4.0) \times 10^{-3}$ | $5.8 \times 10^{-2}$ | fig. 1 |
| $a = 0.55$ | $-(2.4 \pm 4.7) \times 10^{-4}$ | $8.2 \times 10^{-1}$ | |
| $a = 0.76$ | $-(1.7 \pm 2.6) \times 10^{-4}$ | $9.4 \times 10^{-1}$ | |
| $b^{\text{base}} = 0.98$ | 0 | 1 | |
| $b = 0.58$ | $(4.0 \pm 2.0) \times 10^{-2}$ | $2.3 \times 10^{-3}$ | fig. 2 |
| $b = 0.80$ | $(1.4 \pm 1.4) \times 10^{-3}$ | $3.4 \times 10^{-1}$ | |
| $b = 1.07$ | $-(3.4 \pm 3.7) \times 10^{-4}$ | $8.8 \times 10^{-1}$ | |
| $\sigma_{p_T}^{\text{base}} = 0.350$ | 0 | 1 | |
| $\sigma_{p_T} = 0.283$ | $(1.2 \pm 0.8) \times 10^{-2}$ | $1.4 \times 10^{-2}$ | fig. 4 |
| $\sigma_{p_T} = 0.360$ | $-(4.9 \pm 3.1) \times 10^{-4}$ | $9.1 \times 10^{-1}$ | |
| $a^{\text{base}} = 0.68,\ b^{\text{base}} = 0.58$ | 0 | 1 | |
| $a = 0.30,\ b = 0.80$ | $(4.6 \pm 1.3) \times 10^{-2}$ | $5.7 \times 10^{-3}$ | fig. 5 |
| $a = 0.76,\ b = 0.98$ | $(1.2 \pm 0.7) \times 10^{-2}$ | $2.1 \times 10^{-2}$ | |

example, when reweighting the $b$ parameter, a number of baseline values could be chosen and the reweighting technique of this paper can then be used to sample between these distributions to ensure $n_{\text{eff}}$ remains sufficiently large. However, while $n_{\text{eff}}/N$ can indicate when the base distribution differs significantly from the alternate distribution, it cannot capture whether the baseline distribution provides full support for the alternate distribution, *i.e.*, if the alternate distribution enters a region of phase space that is not generated from the baseline distribution. Consequently, table 1 provides the mean event weights, as well as $n_{\text{eff}}/N$, for different values of $a$, $b$, $r_b$, and $\sigma_{p_T}$.

If one compares the mean event weight in table 1 to the distributions in figs. 1 to 5, one can see that the proximity of the mean event weight to unity is a good predictor of the similarity of the distributions.

## 3.2 Timing

A clear benefit of using the reweighting method is that it is universally faster than generating new samples with the alternative parameter values set explicitly. To demonstrate this, we generate a set of $10^2$ samples with $10^3$ events each, using the same PYTHIA 8 settings described above, where we calculate weights for an additional alternative parameter value in each sample. We measure the time it takes to generate each event using a single 2.5 GHz Intel Xeon CPU. Figure 7 shows the arithmetic mean of the time spent to generate a single event as a function of the number of alternative values calculated for Lund parameter $a$. As shown, the marginal cost per additional parameter variation is $\approx 0.04\,\text{ms}$, and it takes $\approx 0.7\,\text{ms}$ to generate an event with 10 alternative values. Since it takes $\approx 0.3\,\text{ms}$ to generate an event with no alternative values, it would take $\approx 3\,\text{ms}$ to generate 10 separate events with the alternative

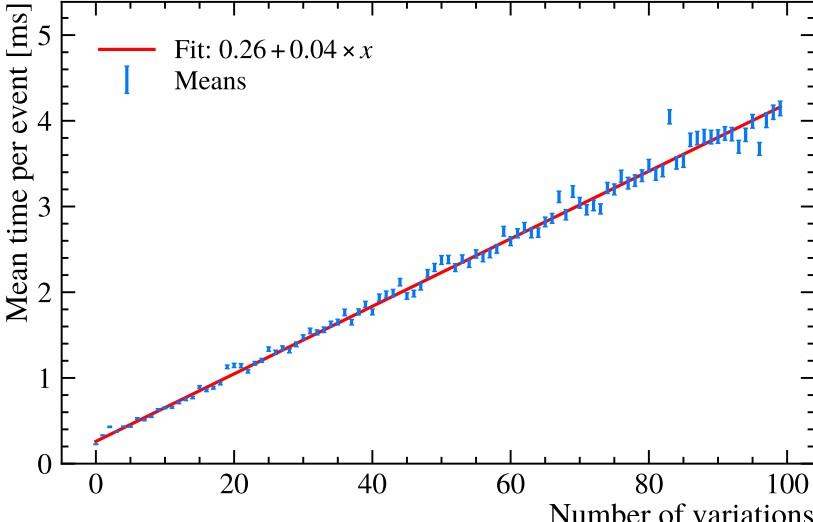

Figure 7: Average time required to generate a single event as a function of the number of alternative parameter values of Lund parameter $a$ calculated during the generation. The error on each point is the standard error of the mean. The amount of time required to generate a single event increases linearly; the best-fit curve is shown in red, and its equation is given in the legend.

values set explicitly, more than 3 times longer than using the modified veto algorithm. These savings vary, depending on the Lund parameter in question, but in all cases, they increase dramatically when one considers detector simulations, which often take $\approx$ 1,000 times longer than the event generation.

## 4 Conclusions

In this study, we have introduced a robust mathematical framework and validated its practical implementation for the fast estimation of hadronization uncertainties in Monte Carlo simulations. By complementing the existing algorithmically efficient uncertainty estimations for the hard matrix-element calculations and parton shower calculations already implemented in PYTHIA 8 [3, 5] and other event generators, our method now offers a rapid estimate of parametric uncertainties for fully hadronized events. Accompanying code is available at gitlab.com/uchep/mlhad-weights-validation, and the reweighting code will be directly incorporated into the next PYTHIA 8 release.

It is important to acknowledge certain limitations of the method: if the parameter variations result in acceptance probability distributions that are far removed from the baseline, the modeling of the new distributions will be poor due to lack of coverage, exemplified in extreme values for the weights. We have found that the deviation of the mean event weight from one is a simple and useful diagnostic tool to identify potential issues; one can also explicitly check that the alternative acceptance probability distribution sufficiently overlaps with the baseline acceptance probability distribution. As long as this coverage condition is met, the presented method provides a practical solution for fast uncertainty estimation in hadronization models, especially in the context of full detector simulations.

# Acknowledgments

We thank L. Gellersen and T. Sjöstrand for careful reading and constructive comments on the manuscript.

**Funding information**     The work was partially completed during the Physics at TeV Colliders Workshop, Les Houches, June 2023. AY, JZ, MS, and TM acknowledge support in part by the DOE grant de-sc0011784 and NSF OAC-2103889. PI and MW are supported in part by NSF OAC-2103889 and NSF-PHY-2209769. SM is supported by the Fermi Research Alliance, LLC under Contract No. DE-AC02-07CH11359 with the U.S. Department of Energy, Office of Science, Office of High Energy Physics. CB acknowledges support from the Knut and Alice Wallenberg foundation, contract number 2017.0036.

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
