# Peer review of "Reweighting Monte Carlo Predictions and Automated Fragmentation Variations in Pythia 8"

_SciPost Physics, doi:SciPost Phys. 16, 134 (2024)_

## Round 2 · Referee Report · Anonymous (Referee 1) · 2023-12-5

Strengths

  1. The authors provide a first implementation and validation of a hadronisation code that can generate alternative event weights for relevant parameter variations, thus greatly reducing the cost of subsequent detector simulation steps in the simulated events pipeline at high-energy colliders such as the LHC. This lays the groundwork to establishing automated on-the-fly hadronisation uncertainties for particle level simulation results, similar to what has been established for on-the-fly hard process and parton shower uncertainties in recent years.

  2. The authors point out potential numerical issues with the ansatz when the baseline and the target probability distributions do not overlap well, as has been observed for shower variations, and attempt to find diagnostics to spot pathological cases. This is also studied in the given validation results.

Weaknesses

  1. Some citations are missing (see "Requested changes").

  2. The criterion of how far the mean is from 1 is not well argued in my opinion, or perhaps I don't understand the point the authors are trying to make here. The actual issue, i.e. the weaker statistical significance due to the wider weight distribution, will of course lead to larger fluctuations of $\mu$ around 1 (as seen from the MC error $\sigma$ given for $(1-\mu)$ for the pathological cases) but then the reliable and more direct measure would be $\sigma$, not $(1-\mu)$, which might be arbitrarily close to 0 given its statistical nature. Alternatively, quoting the effective sample size might be the most natural measure and would at the same time communicate the loss of significance for the alternative weight event sample in a clear way.

Report

This submission reports the first development and application of reweighting methods to a hadronisation model (here, Lund String hadronisation, which is one of two main models in wide use) and the validation of an implementation that enables for the first time the generation of alternative-weight samples for hadronisation uncertainty studies.

Mostly the same methods have been developed for the perturbative parts of the Monte Carlo simulation toolchain (matrix elements, parton showers, matching/merging) between 2011 and 2016 and their use has become a standard for large-scale simulated event sample production at the LHC by ATLAS and CMS and a useful tool for phenomenological studies alike. I expect that the calculation of alternative weights for the hadronisation part will have a similar impact.

Therefore, the submission is a highly relevant contribution. It is of a high quality and definitely worth publishing in SciPost Physics. However, I include a few points in "Requested changes" which were confusing to me and/or that I think could be improved, and I would ask the authors to address them in a minor revision.

Requested changes

  1. In the introduction on page 2, after "efficient methods exist for the hard process and the parton shower", it is in my opinion not sufficient to cite only the VINCIA and PYTHIA reweighting-related publications [3, 4]. As for the hard process, LO reweighting is trivial, but publications that developed NLO reweighting should be added, e.g. [1310.7439] for the reweighting of NLO Monte-Carlo using the Catani-Seymour subtraction. When it comes to the shower (which might include reweighting in the context of matching and merging which should perhaps be mentioned, too), the implementations of the two other general-purpose generators heavily in use at the LHC, in [1605.08256] (HERWIG) and [1606.08753] (SHERPA), should be cited.

  2. The previous point also applies to the use of the same citation group [3, 4] on page 2 after the sentence "The presented method is similar to the one used previously for parton shower uncertainty estimates", i.e. [1605.08256] and [1606.08753] should be cited, too. Here, as you now refer to the reweighting of the veto algorithm, one should also cite [0912.3501], which predates any of the parton shower uncertainty papers by about two years and uses the same "modified veto algorithm" method, albeit to bias shower emissions to generate additional photons. See App. B of [0912.3501].

  3. Since in my (limited) understanding the HERWIG cluster hadronization model is itself an implementation based on ideas from the 80s [Nucl. Phys. B214 (1983) 201, Nucl. Phys. B239 (1984) 349, Nucl. Phys. B288 (1987) 729, Nucl. Phys. B238 (1984) 492], it is unclear why it was picked out as an example, instead of citing also the second implementation of this method in a widely-used general-purpose event generator, i.e. in SHERPA [hep-ph/0311085].

  4. As mentioned in the "Weaknesses" part of the review, I am not convinced by the arguments around eqs. (13)-(15) to put forward the deviation of the mean from unity as the most prominent/straightforward criterion to assess the quality of the reweighting. Isn't it much clearer/more straightforward (and less dependent on random fluctuations, which might send the mean $\mu$ arbitrarily close to unity even if $\sigma$ is large) to use the MC error $\sigma$ of the $w'$ sample itself, or the effective sample size? The eqs. (13)-(15) are not required to establish this even, as it is self-evident that the weight distribution widens by the reweighting. So I would ask the authors to either (i) point out what I have misunderstood and/or (ii) clarify their reasoning in the draft, why the deviation of the mean itself is the relevant criterion here or (iii) just use the greater Monte-Carlo error and/or reduced effective sample size as a criterion and discuss that for the given results. In the latter case, it would be interesting to quote the effective sample size in Tab. 1.

  • validity: top
  • significance: top
  • originality: good
  • clarity: high
  • formatting: perfect
  • grammar: perfect

Author:  Philip Ilten  on 2024-03-12  [id 4359]

(in reply to Report 1 on 2023-12-05)
Category:
remark
answer to question
correction
suggestion for further work

We thank the reviewer for their constructive comments, which we address in detail below. We note that since the submission we have further developed the code which has resulted in some of the distributions and numbers changing (in particular, a bug in how the diquark production was handled was fixed).

A.2 Weaknesses

  1. Thank you for the suggested citations; we have added them.
  2. We agree that we should also quote the effective sample size in addition to the 1 − μ metric, and give further details below about why we think 1 − μ is still a relevant metric.

A.4 Requested changes

  1. Thank you for the suggestions. We have added these citations. For matching and merging, we have included a sentence with citations, although we note that typically these methods are not used in the context of variations or uncertainty estimation. We have tried to cover the major methods (initial methods, POWHEG, MC@NLO, CKKW-L, MLM, MENLOPS, UMEPS, UNLOPS, and FxFx), but since this is a very active area, it is possible that we have missed relevant citations, so please let us know, if you notice any that need adding.
  2. Thank you, we have now included these citations at that point.
  3. This is an oversight on our part. We have added citations here.
  4. We agree that we should include the effective sample size in Table 1, which we have now done. However, we have still included 1 − μ for the following reason. Consider a reweighted distribution that is wider than the underlying base distribution. As you note, the neff will tell us the reduced sample size. However, if our reweighted distribution extends beyond the support of the base distribution, this will not be apparent from neff, and we will be in a scenario where we are missing a substantial portion of our phase space. Here, the 1 − μ distribution is useful because it will now statistically deviate from zero and indicate that we do not have proper support for the reweighted distribution. Additionally, if our reweighting algorithm is simply wrong, we will see a statistical deviation of 1 − μ from zero. Already in the writing of this paper 1 − μ helped us find a number of bugs in our implemented algorithm. We have added an additional paragraph after the discussion of 1 − μ to clarify this.

---

## Round 2 · Referee Report · Anonymous (Referee 2) · 2024-1-3

Strengths

  1. In the manuscript, the authors presented for the first time the possibility of reweighting Monte Carlo hadronisation predictions.

  2. This approach opens up a new avenue to achieve hadronisation uncertainty efficiently (especially if full detector simulation is considered) and can be potentially used for tuning (fitting) hadronisation models.

  3. The manuscript is supplemented with publicly available code (gitlab.com/uchep/mlhad-weights-validation).

Weaknesses

  1. The authors consider a simplified version of the string model (i.e. they reweighted just a few parameters of the model).
    1. It is unclear how this method can be applied when the base function has a domain with zero value.
    2. Some citations are missing (see also the other report)

Report

In the manuscript “Reweighting Monte Carlo Predictions and Automated Fragmentation Variations in Pythia 8” the authors present for the first time a framework to obtain reweighting Monte Carlo hadronisation predictions. This approach opens up new possibilities for efficiently obtaining hadronisation uncertainties (especially if a full detector simulation is considered) and can potentially be used to tune (fit) hadronisation models. The results presented are, in my opinion, very interesting. However, before the article is published, I have some comments/questions (see requested changes), the answers to which I think would help to further improve the article.

Requested changes

Requested changes (in order of appearance in the text):

  1. In the Introduction, the authors write that the proposed approach can, for example, be applied to a multi-particle interaction model. This does not seem obvious to me. It is also not clear to me how the approach could be applied for colour reconnection - which could be considered a part of hadronization or MPI. Could the authors please elaborate on this more?

  2. It would be good to add references to more modern versions of the software and add references to Parton Shower/Generators uncertainty studies of other groups.

3a. The authors consider a simplified version of the string model. For example, they neglect to re-weight the flavour parameters of the model. What is the reason for this? The string model has many more parameters (for example, Monash Tune, which the authors use as the default setting for the Lund model, had more than 20 hadronisation parameters tuned). How would the methods work for such a large number of parameters? What are the potential problems in such a more realistic situation?

3b. Related to this question is the problem described at the bottom of page 4 concerning low-mass strings. The authors write openly about this problem, but it is not clear what exact limitations this problem brings to the estimation of hadronisation uncertainty using the proposed method.

  1. Some of the parameters of the string model are discreet. How the method can be applied to the discreet parameters?

  2. In section 3.1 the authors write: “This agreement breaks down, if the Lund fragmentation function for the alternative parameter values is large in a range where the Lund fragmentation function approaches zero for the baseline parameter values, as shown in fig. 2 (bottom left) and fig. 3 (bottom right). The reweighting then requires large weights and samples the phase space poorly.” It is even worse in the case when the base function has a domain where it is zero and the function for the alternative parameter is non-zero. Clearly, in that situation, the method can not be applied. This appears to be a serious limitation of the method. It would be interesting to see what solutions the authors propose in such a situation.

  3. In Fig. 2 in the bottom panel for the case of b=0.58 there is a large error bar for w’ line between charge multiplicity 30-35. What is the reason for that?

  4. In perturbation calculations, a variation of the renormalisation and factorisation scales is usually used as a rough estimate of uncertainty. What variations in the parameter of the Lund model would the authors suggest to estimate the uncertainties associated with it?

In summary, I would like to say that I think the document is very interesting. However, there are still some issues which have to be addressed before the publication. Therefore, I recommend that the author addresses the points raised above before resubmitting their paper.

  • validity: high
  • significance: high
  • originality: high
  • clarity: high
  • formatting: excellent
  • grammar: excellent

Author:  Philip Ilten  on 2024-03-12  [id 4360]

(in reply to Report 2 on 2024-01-03)
Category:
remark
answer to question
correction
suggestion for further work

We thank the reviewer for their constructive comments. We address them in detail below. We note that since submission we have further developed the code which has resulted in some of the distributions and numbers changing (in particular, a bug in how the diquark production was handled was fixed).

B.2 Weaknesses

  1. We should clarify here that the string model itself is not simplified, this is implemented for the full Pythia 8 string model. However, we have only implemented reweighting for a subset of parameters for the Pythia 8 hadronization model. While this is a subset of parameters, these are the primary kinematic parameters that are used in any Pythia 8 tune, and consequently are the most relevant for the majority of simulations performed with Pythia 8. Additionally, this is a proof-of-concept and the same reweighting techniques can be used for second-order parameters, e.g., the use of the non-standard Lund ansatz. There is one exception, which is the reweighting of flavor parameters, rather than kinematic parameters. Here, the Pythia 8 flavor selector does not use an accept-reject algorithm, and so a different approach needs to be taken which is currently being investigated, but is outside the scope of this manuscript.
  2. Indeed, our method cannot be applied when the base function does not have coverage; this is a weakness for any reweighting technique. This is the reason why we introduce the 1 − μ metric, as this indicates to the user whether the base function has entered such a regime. We have included an extended discussion of this in the manuscript, where we give an example of how 1 − μ can be used to identify insufficient support of the base distribution, and how then additional base distributions can be used, with the reweighting used to span these base distributions.
  3. We have now addressed these missing citations.

B.4 Requested changes

  1. In the multi-parton interaction (MPI) model of Pythia 8, additional 2 → 2 scatterings are produced, interleaved with the parton shower. These additional interactions produce further strings which are automatically handled by our method because the weight for an event is the product of not just the weights for string breaks for a single string, but for the weights of string breaks of all strings in the event. It is important to note that, if the number of strings for an event becomes large, then the distribution of weights will begin to have a very large spread, thus reducing the effective sample size, and reducing the efficacy of our method. For color reconnection, the strings are just rearranged, and so our method is still applicable. Since the MPI model is implemented with parton-shower-like veto algorithm the variations of MPI parameters could be handled with the same reweighting method described here (with obvious modifications). Reweighting for different color reconnection models, however, is an entirely different problem which is outside the scope of this manuscript. We have added a sentence in the introduction describing this.
  2. Thank you for the suggestion, we have added additional references. 3a. As we note above, we are not working with a simplified version of the string model; all plots made for this manuscript use the full Pythia 8 hadronization model. However, we do only explore reweighting for the first-order kinematic parameters. Additional kinematic parameters, such as a non-standard Lund ansatz could also have the same method applied, but this is more a question of Pythia 8 development rather than the method itself. In the Monash tune, 18 hadronization related parameters are considered: 12 of these are flavor parameters and 6 are kinematic parameters. Of the 6 kinematic parameters relevant to this manuscript, we have introduced 5 of them as reweightable (the exception is StringZ:aExtraDiquark), so we would argue this is already a relatively complete implementation. Reweighting flavor-based parameters is very different from the kinematic parameters as the Pythia 8 flavor selector does not use the same accept-reject algorithm as for the string kinematics. An independent study is underway on how to reweight flavor parameters, but this is outside the scope of this manuscript. We have added additional text to the manuscript to clarify this point. 3b. This is an important point; thank you for bringing it up. We have looked further into the issue of the final combination of the low mass string, and have updated the text accordingly. The final combination rejection rate depends only on the preceding string kinematics, and not on any of the kinematic parameters. The b parameter is used to break the symmetry of the system, but is not used in determining whether the entire set of string breaks should be accepted or rejected. Consequently, it is possible to account for this final step by simply including the weights of the rejected strings in addition to the accepted strings.
  3. Some of the discrete parameters can be handled in the same way. Consider for example the parameter StringZ:useNonstandardC which uses, f (z) = 1/z^(1+r_Qb_Qm_Q^2 )*(1 − z)^a_Q exp(−b_Qm_T^2 /z ). The same methodology can be applied here. However, some discrete parameters cannot be handled in the same way, as they change the model. An example would be StringPT:thermalModel. In this case a new or updated reweighting scheme would need to be introduced for the alternate model.
  4. Thank you for the suggestion. In the discussion around 1 − μ in the manuscript we have added some additional text. Specifically, we would suggest that the 1 − μ metric is closely monitored. Once it is clear that this statistically deviates from zero, then a new base distribution should be generated. In this way, the reweighted distributions finely span the coarse separation of the base distributions. We would foresee that for a given tuning exercise, depending upon the necessary granularity, only a few base distributions would need to be generated. If the reweighting is used for uncertainty estimation, then we expect that for most cases a single base distribution should be sufficient.
  5. This is a statistical fluctuation. We have confirmed that when we generate with ten times more statistics that this large error bar disappears.
  6. In general, we think this is beyond the scope of the manuscript, in that we have developed a method to evaluate the uncertainties, but do not aim to determine the range of the parameters to properly encompass the uncertainties. Analogously, the Hessian method for PDF uncertainty is independent of the eigensets for a given PDF. Of course, this technique could be used in a dedicated tuning effort which could then estimate the range for the parameters, in a similar fashion to what has already been done with eigentunes. We do note that there are a number of papers that do give guidance on variations for some of the Pythia 8 parameters. For example, in arXiv:1610.08328 the value for b is estimated as 1.04+0.01−0.02, while a range of [0.2, 2] was considered. For such a small parameter range, the reweighting method is expected to perform very well with both proper support and a high neff.

---

## Round 3 · Referee Report · Anonymous (Referee 1) · 2024-3-25

Report

This is a follow-up report to my report submitted at 2023-12-5.

The draft has been revised to meet all requested changes, thus removing the identified weaknesses.

I particularly welcome the improved discussion of the quality criteria, i.e. the discussion on $1-\mu$, and the addition of the effective sample size.

When I noted that the "reweighting in the context of matching and merging [...] should perhaps be mentioned", I was rather pointing to the existence of on-the-fly variation methods that even work for matched and merged simulations [arXiv:1606.08753], not so much the non-variational application of reweighting in matching/merging. However, that is a very minor quibble, and I explicitly do not request another change on the basis of that, especially because this is somehow implicit in the preceding formulation "efficient reweighting methods exist for the hard process and the parton shower" (as matching/merging is just a combination of the hard process and the parton shower), and all the proper citations are already given for that.

Hence, I suggest this revision for publication without further changes.

---

## Round 3 · Referee Report · Anonymous (Referee 2) · 2024-4-10

Strengths

I thank the authors for answering my questions from my first report and I am pleased that the code has been further improved and that the production of diquarks is now correctly accounted for.

Weaknesses

The authors have responded to all the weaknesses contained in my previous report.

Report

I propose that the article be accepted for publication.

Recommendation

Publish (easily meets expectations and criteria for this Journal; among top 50%)

---

## Round 3 · Author Response

We thank the reviewers for the detailed reading of our manuscript. We have updated our manuscript with a number of changes based on their suggestions.

---

## Round 3 · List of Changes

• Added missing references, as suggested by the referees.
  • Clarified how the reweighting method outlined can be used with MPI switched on, as well as how the general methods are applicable to varying MPI parameters.
  • Included an extended discussion on using 1 - mu as well as the number of effective events, and why both metrics are necessary when performing variations.
  • The effective number of events has now been included in Table 1.
  • Fixed a bug in the code where the diquark production was not being handled correctly, which resulted in a small change in the distributions.
  • Corrected our discussion about how the final string end is joined into two hadrons. Specifically, the weights of the strings which fail this final joining need to be included in the overall weight of the event.

---

## Editorial Decision

published